# CANITA: Faster Rates for Distributed Convex Optimization with Communication Compression

**Zhize Li**
KAUST
zhize.li@kaust.edu.sa

**Peter Richtárik**
KAUST
peter.richtarik@kaust.edu.sa

## Abstract

Due to the high communication cost in distributed and federated learning, methods relying on compressed communication are becoming increasingly popular. Besides, the best theoretically and practically performing gradient-type methods invariably rely on some form of acceleration/momentum to reduce the number of communications (faster convergence), e.g., Nesterov's accelerated gradient descent [31, 32] and Adam [14]. In order to combine the benefits of communication compression and convergence acceleration, we propose a *compressed and accelerated* gradient method based on ANITA [20] for distributed optimization, which we call CANITA. Our CANITA achieves the *first accelerated rate* $O\left(\sqrt{\left(1+\sqrt{\frac{\omega^3}{n}}\right)\frac{L}{\epsilon}} + \omega\left(\frac{1}{\epsilon}\right)^{\frac{1}{3}}\right)$, which improves upon the state-of-the-art non-accelerated rate $O\left((1+\frac{\omega}{n})\frac{L}{\epsilon} + \frac{\omega^2+\omega}{\omega+n}\frac{1}{\epsilon}\right)$ of DIANA [12] for distributed general convex problems, where $\epsilon$ is the target error, $L$ is the smooth parameter of the objective, $n$ is the number of machines/devices, and $\omega$ is the compression parameter (larger $\omega$ means more compression can be applied, and no compression implies $\omega = 0$). Our results show that as long as the number of devices $n$ is large (often true in distributed/federated learning), or the compression $\omega$ is not very high, CANITA achieves the faster convergence rate $O\left(\sqrt{\frac{L}{\epsilon}}\right)$, i.e., the number of communication rounds is $O\left(\sqrt{\frac{L}{\epsilon}}\right)$ (vs. $O\left(\frac{L}{\epsilon}\right)$ achieved by previous works). As a result, CANITA enjoys the advantages of both compression (compressed communication in each round) and acceleration (much fewer communication rounds).

## 1 Introduction

With the proliferation of edge devices, such as mobile phones, wearables and smart home appliances, comes an increase in the amount of data rich in potential information which can be mined for the benefit of humankind. One of the approaches of turning the raw data into information is via federated learning [15, 29], where typically a single global supervised model is trained in a massively distributed manner over a network of heterogeneous devices.

Training supervised distributed/federated learning models is typically performed by solving an optimization problem of the form

$$\min_{x\in\mathbb{R}^d}\left\{f(x) := \frac{1}{n}\sum_{i=1}^{n}f_i(x)\right\}, \tag{1}$$

where $n$ denotes the number of devices/machines/workers/clients, and $f_i : \mathbb{R}^d \to \mathbb{R}$ is a loss function associated with the data stored on device $i$. We will write

$$x^* := \arg\min_{x\in\mathbb{R}^d}f(x).$$

35th Conference on Neural Information Processing Systems (NeurIPS 2021).

If more than one minimizer exist, $x^*$ denotes an arbitrary but fixed solution. We will rely on the solution concept captured in the following definition:

**Definition 1** *A random vector $\widehat{x} \in \mathbb{R}^d$ is called an $\epsilon$-solution of the distributed problem (1) if*

$$\mathbb{E}[f(\widehat{x})] - f(x^*) \leq \epsilon,$$

*where the expectation is with respect to the randomness inherent in the algorithm used to produce $\widehat{x}$.*

In distributed and federated learning problems of the form (1), communication of messages across the network typically forms the key bottleneck of the training system. In the modern practice of supervised learning in general and deep learning in particular, this is exacerbated by the reliance on massive models described by millions or even billions of parameters. For these reasons, it is very important to devise novel and more efficient training algorithms capable of decreasing the overall communication cost, which can be formalized as the product of the number of communication rounds necessary to train a model of sufficient quality, and the computation and communication cost associated with a typical communication round.

## 1.1 Methods with compressed communication

One of the most common strategies for improving communication complexity is *communication compression* [37, 1, 40, 8, 30, 9, 26, 24]. This strategy is based on the reduction of the size of communicated messages via the application of a suitably chosen lossy compression mechanism, saving precious time spent in each communication round, and hoping that this will not increase the total number of communication rounds.

Several recent theoretical results suggest that by combining an appropriate (randomized) compression operator with a suitably designed gradient-type method, one can obtain improvement in the total communication complexity over comparable baselines not performing any compression. For instance, this is the case for distributed compressed gradient descent (CGD) [1, 13, 8, 24], and distributed CGD methods which employ variance reduction to tame the variance introduced by compression [7, 30, 9, 24, 6].

## 1.2 Methods with acceleration

The acceleration/momentum of gradient-type methods is widely-studied in standard optimization problems, which aims to achieve faster convergence rates (fewer communication rounds) [33, 31, 32, 17, 28, 2, 18, 16, 23, 20]. Deep learning practitioners typically rely on Adam [14], or one of its many variants, which besides other tricks also adopts momentum. In particular, ANITA [20] obtains the current state-of-the-art convergence results for convex optimization. In this paper, we will adopt the acceleration from ANITA [20] to the distributed setting with compression.

## 1.3 Can communication compression and acceleration be combined?

Encouraged by the recent theoretical success of communication compression, and the widespread success of accelerated methods, in this paper we seek to further enhance CGD methods with acceleration/momentum, with the aim to obtain provable improvements in overall communication complexity.

> *Can distributed gradient-type methods theoretically benefit from the combination of gradient compression and acceleration/momentum? To the best of our knowledge, no such results exist in the general convex regime, and in this paper we close this gap by designing a method that can provably enjoy the advantages of both compression (compressed communication in each round) and acceleration (much fewer communication rounds).*

While there is abundance of research studying communication compression and acceleration in isolation, there is very limited work on the combination of both approaches. The first successful combination of gradient compression and acceleration/momentum was recently achieved by the ADIANA method of Li et al. [26]. However, Li et al. [26] only provide theoretical results for strongly convex problems, and their method is not applicable to (general) convex problems. So, one needs to

Table 1: Convergence rates for finding an $\epsilon$-solution $\mathbb{E}[f(x^T)] - f(x^*) \leq \epsilon$ of distributed problem (1)

| Algorithms | Strongly convex [1] | General convex | Remark |
|---|---|---|---|
| QSGD [1] | — | $O\left(\frac{L}{\epsilon} + \frac{\omega G^2}{n}\frac{1}{\epsilon^2}\right)$ [2] | ✓compression
✗ acceleration |
| DIANA [30] | $O\left(\left(\left(1+\frac{\omega}{n}\right)\kappa+\omega\right)\log\frac{1}{\epsilon}\right)$ | — | ✓compression
✗ acceleration |
| DIANA [9] | $O\left(\left(\left(1+\frac{\omega}{n}\right)\kappa+\omega\right)\log\frac{1}{\epsilon}\right)$ | $O\left(\left(1+\frac{\omega}{n}\right)\frac{L}{\epsilon}+\frac{\omega}{\epsilon}\right)$ | ✓compression
✗ acceleration |
| DIANA [12] | — | $O\left(\left(1+\frac{\omega}{n}\right)\frac{L}{\epsilon}+\frac{\omega^2+\omega}{\omega+n}\frac{1}{\epsilon}\right)$ | ✓compression
✗ acceleration |
| ADIANA [26] | $O\left(\left(\sqrt{\kappa}+\sqrt{\left(\frac{\omega}{n}+\sqrt{\frac{\omega}{n}}\right)\omega\kappa}+\omega\right)\log\frac{1}{\epsilon}\right)$ | — | ✓compression
✓acceleration |
| CANITA
(this paper) | — | $O\left(\sqrt{\left(1+\sqrt{\frac{\omega^3}{n}}\right)\frac{L}{\epsilon}}+\omega\left(\frac{1}{\epsilon}\right)^{\frac{1}{3}}\right)$ | ✓compression
✓acceleration |

both design a new method to handle the convex case, and perform its analysis. A-priori, it is not clear at all what approach would work.

To the best of our knowledge, besides the initial work [26], we are only aware of two other works for addressing this question [41, 34]. However, both these works still only focus on the simpler and less practically relevant *strongly convex* setting. Thus, this line of research is still largely unexplored. For instance, the well-known logistic regression problem is convex but not strongly convex. Finally, even if a problem is strongly convex, the modulus of strong convexity is typically not known, or hard to estimate properly.

## 2   Summary of Contributions

In this paper we propose and analyze an accelerated gradient method with compressed communication, which we call CANITA (described in Algorithm 1), for solving distributed *general convex* optimization problems of the form (1). In particular, CANITA can loosely be seen as a combination of the accelerated gradient method ANITA of [20], and the variance-reduced compressed gradient method DIANA of [30]. Ours is the first work provably combining the benefits of communication compression and acceleration in the general convex regime.

### 2.1   First accelerated rate for compressed gradient methods in the convex regime

For general convex problems, CANITA is the first compressed communication gradient method with an *accelerated rate*. In particular, our CANITA solves the distributed problem (1) in

$$O\left(\sqrt{\left(1+\sqrt{\frac{\omega^3}{n}}\right)\frac{L}{\epsilon}}+\omega\left(\frac{1}{\epsilon}\right)^{\frac{1}{3}}\right)$$

communication rounds, which improves upon the current state-of-the-art result

$$O\left(\left(1+\frac{\omega}{n}\right)\frac{L}{\epsilon}+\frac{\omega^2+n}{\omega+n}\frac{1}{\epsilon}\right)$$

achieved by the DIANA method [12]. See Table 1 for more comparisons.

Let us now illustrate the improvements coming from this new bound on an example with concrete numerical values. Let the compression ratio be $10\%$ (the size of compressed message is $0.1 \cdot d$, where $d$ is the size of the uncompressed message). If random sparsification or quantization is used to achieve this, then $\omega \approx 10$ (see Section 3.1). Further, if the number of devices/machines is $n = 10^6$,

---

[1]In this strongly convex column, $\kappa := \frac{L}{\mu}$ denotes the condition number, where $L$ is the smooth parameter and $\mu > 0$ is the strong convexity parameter.

[2]Here QSGD [1] needs an additional bounded gradient assumption, i.e., $\|\nabla f_i(x)\|^2 \leq G^2, \forall i \in [n], x \in \mathbb{R}^d$.

and the target error tolerance is $\epsilon = 10^{-6}$, then the number of communication rounds of our CANITA method is $O(10^3)$, while the number of communication rounds of the previous state-of-the-art method DIANA [12] is $O(10^6)$, i.e., $O\left(\sqrt{\frac{L}{\epsilon}}\right)$ vs. $O(\frac{L}{\epsilon})$. *This is an improvement of three orders of magnitude.*

Moreover, the numerical experiments in Section 6 indeed show that the performance of our CANITA is much better than previous non-accelerated compressed methods (QSGD and DIANA), corroborating the theoretical results (see Table 1) and confirming the practical superiority of our accelerated CANITA method.

## 2.2 Accelerated rate with limited compression for free

For strongly convex problems, Li et al. [26] showed that if the number of devices/machines $n$ is large, or the compression variance parameter $\omega$ is not very high ($\omega \leq n^{1/3}$), then their ADIANA method enjoys the benefits of both compression and acceleration (i.e., $\sqrt{\kappa} \log \frac{1}{\epsilon}$ of ADIANA vs. $\kappa \log \frac{1}{\epsilon}$ of previous works).

In this paper, we consider the general convex setting and show that the proposed CANITA also enjoys the benefits of both compression and acceleration. Similarly, if $\omega \leq n^{1/3}$ (i.e., many devices, or limited compression variance), CANITA achieves the accelerated rate $\sqrt{\frac{L}{\epsilon}}$ vs. $\frac{L}{\epsilon}$ of previous works. This means that the compression does not hurt the accelerated rate at all. Note that the second term $\left(\frac{1}{\epsilon}\right)^{\frac{1}{3}}$ is of a lower order compared with the first term $\sqrt{\frac{L}{\epsilon}}$.

## 2.3 Novel proof technique

The proof behind the analysis of CANITA is significantly different from that of ADIANA [26], which critically relies on strong convexity. Moreover, the theoretical rate in the strongly convex case is linear $O(\log \frac{1}{\epsilon})$, while it is sublinear $O(\frac{1}{\epsilon})$ or $O\left(\sqrt{\frac{1}{\epsilon}}\right)$ (accelerated) in the general convex case. We hope that our novel analysis can provide new insights and shed light on future work.

# 3 Preliminaries

Let $[n]$ denote the set $\{1, 2, \cdots, n\}$ and $\| \cdot \|$ denote the Euclidean norm for a vector and the spectral norm for a matrix. Let $\langle u, v \rangle$ denote the standard Euclidean inner product of two vectors $u$ and $v$. We use $O(\cdot)$ and $\Omega(\cdot)$ to hide the absolute constants.

## 3.1 Assumptions about the compression operators

We now introduce the notion of a randomized *compression operator* which we use to compress the gradients to save on communication. We rely on a standard class of unbiased compressors (see Definition 2) that was used in the context of distributed gradient methods before [1, 13, 9, 24, 26].

**Definition 2 (Compression operator)** *A randomized map $\mathcal{C} : \mathbb{R}^d \mapsto \mathbb{R}^d$ is an $\omega$-compression operator if*
$$\mathbb{E}\left[\mathcal{C}(x)\right] = x, \qquad \mathbb{E}\left[\|\mathcal{C}(x) - x\|^2\right] \leq \omega \|x\|^2, \qquad \forall x \in \mathbb{R}^d. \tag{2}$$
*In particular, no compression ($\mathcal{C}(x) \equiv x$) implies $\omega = 0$.*

It is well known that the conditions (2) are satisfied by many practically useful compression operators (see Table 1 in [3, 36]). For illustration purposes, we now present a couple canonical examples: sparsification and quantization.

**Example 1 (Random sparsification).** Given $x \in \mathbb{R}^d$, the random-$k$ sparsification operator is defined by
$$\mathcal{C}(x) := \frac{d}{k} \cdot (\xi_k \odot x),$$
where $\odot$ denotes the Hadamard (element-wise) product and $\xi_k \in \{0, 1\}^d$ is a uniformly random binary vector with $k$ nonzero entries ($\|\xi_k\|_0 = k$). This random-$k$ sparsification operator $\mathcal{C}$ satisfies

(2) with $\omega = \frac{d}{k} - 1$. By setting $k = d$, this reduces to the identity compressor, whose variance is obviously zero: $\omega = 0$.

**Example 2 (Random quantization).**  Given $x \in \mathbb{R}^d$, the $(p, s)$-quantization operator is defined by

$$\mathcal{C}(x) := \text{sign}(x) \cdot \|x\|_p \cdot \frac{1}{s} \cdot \xi_s,$$

where $p, s \geq 1$ are integers, and $\xi_s \in \mathbb{R}^d$ is a random vector with $i$-th element

$$\xi_s(i) := \begin{cases} l + 1, & \text{with probability } \frac{|x_i|}{\|x\|_p} s - l, \\ l, & \text{otherwise.} \end{cases}$$

The level $l$ satisfies $\frac{|x_i|}{\|x\|_p} \in [\frac{l}{s}, \frac{l+1}{s}]$. The probability is chosen so that $\mathbb{E}[\xi_s(i)] = \frac{|x_i|}{\|x\|_p} s$. This $(p, s)$-quantization operator $\mathcal{C}$ satisfies (2) with $\omega = 2 + \frac{d^{1/p} + d^{1/2}}{s}$. In particular, QSGD [1] used $p = 2$ (i.e., $(2, s)$-quantization) and proved that the expected sparsity of $\mathcal{C}(x)$ is $\mathbb{E}[\|\mathcal{C}(x)\|_0] = O(s(s + \sqrt{d}))$.

### 3.2   Assumptions about the functions

Throughout the paper, we assume that the functions $f_i$ are convex and have Lipschitz continuous gradient.

**Assumption 1** *Functions $f_i : \mathbb{R}^d \to \mathbb{R}$ are convex, differentiable, and $L$-smooth. The last condition means that there exists a constant $L > 0$ such that for all $i \in [n]$ we have*

$$\|\nabla f_i(x) - \nabla f_i(y)\| \leq L \|x - y\|, \qquad \forall x, y \in \mathbb{R}^d. \tag{3}$$

It is easy to see that the objective $f(x) = \frac{1}{n} \sum_{i=1}^n f_i(x)$ in (1) satisfies (3) provided that the constituent functions $\{f_i\}$ do.

## 4   The CANITA Algorithm

In this section, we describe our method, for which we coin the name CANITA, designed for solving problem (1), which is of importance in distributed and federated learning, and contrast it to the most closely related methods ANITA [20], DIANA [30] and ADIANA [26].

---

**Algorithm 1** Distributed compressed accelerated ANITA method (CANITA)

---

**Input:** initial point $x^0 \in \mathbb{R}^d$, initial shift vectors $h_1^0, \ldots, h_n^0 \in \mathbb{R}^d$, probabilities $\{p_t\}$, and positive stepsizes $\{\alpha_t\}, \{\eta_t\}, \{\theta_t\}$
1: **Initialize:** $w^0 = z^0 = x^0$ and $h^0 = \frac{1}{n} \sum_{i=1}^n h_i^0$
2: **for** $t = 0, 1, 2, \ldots$ **do**
3:    $y^t = \theta_t x^t + (1 - \theta_t) w^t$
4:    **for all machines** $i = 1, 2, \ldots, n$ **do in parallel**
5:       Compress the shifted local gradient $\mathcal{C}_i^t(\nabla f_i(y^t) - h_i^t)$ and send the result to the server
6:       Update the local shift $h_i^{t+1} = h_i^t + \alpha_t \mathcal{C}_i^t(\nabla f_i(w^t) - h_i^t)$
7:    **end for**
8:    Aggregate received compressed local gradient information:

$$g^t = h^t + \frac{1}{n} \sum_{i=1}^n \mathcal{C}_i^t(\nabla f_i(y^t) - h_i^t) \qquad \bullet \text{ Compute gradient estimator}$$

$$h^{t+1} = h^t + \alpha_t \frac{1}{n} \sum_{i=1}^n \mathcal{C}_i^t(\nabla f_i(w^t) - h_i^t) \qquad \bullet \text{ Maintain the average of local shifts}$$

9:    Perform update step:
      $x^{t+1} = x^t - \frac{\eta_t}{\theta_t} g^t$
10:    $z^{t+1} = \theta_t x^{t+1} + (1 - \theta_t) w^t$
11:    $w^{t+1} = \begin{cases} z^{t+1}, & \text{with probability } p_t \\ w^t, & \text{with probability } 1 - p_t \end{cases}$
12: **end for**

---

### 4.1 CANITA: description of the method

Our proposed method CANITA, formally described in Algorithm 1, is an accelerated gradient method supporting compressed communication. It is the first method combing the benefits of acceleration and compression in the general convex regime (without strong convexity).

In each round $t$, each machine computes its local gradient (e.g., $\nabla f_i(y^t)$) and then a shifted version is compressed and sent to the server (See Line 5 of Algorithm 1). The local shifts $h_i^t$ are adaptively changing throughout the iterative process (Line 6), and have the role of reducing the variance introduced by compression $\mathcal{C}(\cdot)$. If no compression is used, we may simply set the shifts to be $h_i^t = 0$ for all $i, t$. The server subsequently aggregates all received messages to obtain the gradient estimator $g^t$ and maintain the average of local shifts $h^{t+1}$ (Line 8), and then perform gradient update step (Line 9) and update momentum sequences (Line 10 and 3). Besides, the last Line 11 adopts a randomized update rule for the auxiliary vectors $w^t$ which simplifies the algorithm and analysis, resembling the workings of the loopless SVRG method used in [16, 20].

### 4.2 CANITA vs existing methods

CANITA can be loosely seen as a combination of the accelerated gradient method ANITA of [20], and the variance-reduced compressed gradient method DIANA of [30]. In particular, CANITA uses momentum/acceleration steps (see Line 3 and 10 of Algorithm 1) inspired by those of ANITA [20], and adopts the shifted compression framework for each machine (see Line 5 and 6 of Algorithm 1) as in the DIANA method [30].

> *We prove that* CANITA *enjoys the benefits of both methods simultaneously, i.e., convergence acceleration of* ANITA *and gradient compression of* DIANA.

Although CANITA can conceptually be seen as combination of ANITA [20] and DIANA [30, 9, 12] from an algorithmic perspective, the analysis of CANITA is entirely different. Let us now briefly outline some of the main differences.

- For example, compared with ANITA [20], CANITA needs to deal with the extra compression of shifted local gradients in the distributed network. Thus, the obtained gradient estimator $g^k$ in Line 8 of Algorithm 1 is substantially different and more complicated than the one in ANITA, which necessitates a novel proof technique.

- Compared with DIANA [30, 9, 12], the extra momentum steps in Line 3 and 10 of Algorithm 1 make the analysis of CANITA more complicated than that of DIANA. We obtain the accelerated rate $O\left(\sqrt{\frac{L}{\epsilon}}\right)$ rather than the non-accelerated rate $O(\frac{L}{\epsilon})$ of DIANA, and this is impossible without a substantially different proof technique.

- Compared with the accelerated DIANA method ADIANA of [26], the analysis of CANITA is also substantially different since CANITA cannot exploit the strong convexity assumed therein.

Finally, please refer to Section 2 where we summarize our contributions for additional discussions.

## 5 Convergence Results for the CANITA Algorithm

In this section, we provide convergence results for CANITA (Algorithm 1). In order to simplify the expressions appearing in our main result (see Theorem 1 in Section 5.1) and in the lemmas needed to prove it (see Appendix A), it will be convenient to let

$$F^t := f(w^t) - f(x^*), \qquad H^t := \frac{1}{n}\sum_{i=1}^{n}\|\nabla f_i(w^t) - h_i^t\|^2, \qquad D^t := \frac{1}{2}\|x^t - x^*\|^2. \quad (4)$$

### 5.1 Generic convergence result

We first present the main convergence theorem of CANITA for solving the distributed optimization problem (1) in the general convex regime.

**Theorem 1** *Suppose that Assumption 1 holds and the compression operators $\{\mathcal{C}_i^t\}$ used in Algorithm 1 satisfy (2) of Definition 2. For any two positive sequences $\{\beta_t\}$ and $\{\gamma_t\}$ such that the probabilities $\{p_t\}$ and positive stepsizes $\{\alpha_t\}, \{\eta_t\}, \{\theta_t\}$ of Algorithm 1 satisfy the following relations*

$$\alpha_t \leq \frac{1}{1+\omega}, \qquad \eta_t \leq \frac{1}{L\left(1 + \beta_t + 4p_t\gamma_t\left(1 + \frac{2p_t}{\alpha_t}\right)\right)} \tag{5}$$

*for all $t \geq 0$, and*

$$\frac{2\omega}{\beta_t n} + 4p_t\gamma_t\left(1 + \frac{2p_t}{\alpha_t}\right) \leq 1 - \theta_t, \quad \frac{(1-p_t\theta_t)\eta_t}{p_t\theta_t^2} \leq \frac{\eta_{t-1}}{p_{t-1}\theta_{t-1}^2}, \quad \left(\frac{\omega}{\beta_t n} + \left(1 - \frac{\alpha_t}{2}\right)\gamma_t\right)\frac{\eta_t}{\theta_t^2} \leq \frac{\gamma_{t-1}\eta_{t-1}}{\theta_{t-1}^2} \tag{6}$$

*for all $t \geq 1$. Then the sequences $\{x^t, w^t, h_i^t\}$ of* CANITA *(Algorithm 1) for all $t \geq 0$ satisfy the inequality*

$$\mathbb{E}\left[F^{t+1} + \frac{\gamma_t p_t}{L}H^{t+1}\right] \leq \frac{\theta_t^2 p_t}{\eta_t}\left(\frac{(1-\theta_0 p_0)\eta_0}{\theta_0^2 p_0}F^0 + \left(\frac{\omega}{\beta_0 n} + \left(1 - \frac{\alpha_0}{2}\right)\gamma_0\right)\frac{\eta_0}{\theta_0^2 L}H^0 + D^0\right), \tag{7}$$

*where the quantities $F^t, H^t, D^t$ are defined in* (4).

The detailed proof of Theorem 1 which relies on six lemmas is provided in Appendix A. In particular, the proof simply follows from the key Lemma 6 (see Appendix A.2), while Lemma 6 closely relies on previous five Lemmas 1–5 (see Appendix C.6). Note that all proofs for these six lemmas are deferred to Appendix C.

As we shall see in detail in Section 5.2, the sequences $\beta_t, \gamma_t, p_t$ and $\alpha_t$ can be fixed to some constants.[3] However, the relaxation parameter $\theta_t$ needs to be decreasing and the stepsize $\eta_t$ may be increasing until a certain threshold. In particular, we choose

$$\beta_t \equiv c_1, \quad \gamma_t \equiv c_2, \quad p_t \equiv c_3, \quad \alpha_t \equiv c_4, \quad \theta_t = \frac{c_5}{t+c_6}, \quad \eta_t = \min\left\{\left(1 + \frac{1}{t+c_7}\right)\eta_{t-1}, \frac{1}{c_8 L}\right\}, \tag{8}$$

where the constants $\{c_i\}$ may depend on the compression parameter $\omega$ and the number of devices/machines $n$. As a result, the right hand side of (7) will be of the order $O\left(\frac{L}{t^2}\right)$, which indicates an *accelerated* rate. Hence, in order to find an $\epsilon$-solution of problem (1), i.e., vector $w^{T+1}$ such that

$$\mathbb{E}\left[f(w^{T+1}) - f(x^*)\right] \overset{(4)}{:=} \mathbb{E}\left[F^{T+1}\right] \leq \epsilon, \tag{9}$$

the number of communication rounds of CANITA (Algorithm 1) is at most $T = O\left(\sqrt{\frac{L}{\epsilon}}\right)$.

While the above rate has an accelerated dependence on $\epsilon$, it will be crucial to study the omitted constants $\{c_i\}$ (see (8)), and in particular their dependence on the compression parameter $\omega$ and the number of devices/machines $n$. As expected, for any fixed target error $\epsilon > 0$, the number of communication rounds $T$ (sufficient to guarantee that (9) holds) may grow with increasing levels of compression, i.e., with increasing $\omega$. However, at the same time, the communication cost in each round decreases with $\omega$. It is easy to see that this trade-off benefits compression. In particular, as we mention in Section 2, if the number of devices $n$ is large, or the compression variance $\omega$ is not very high, then compression does not hurt the accelerated rate of communication rounds at all.

## 5.2 Detailed convergence result

We now formulate a concrete Theorem 2 from Theorem 1 which leads to a detailed convergence result for CANITA (Algorithm 1) by specifying the choice of the parameters $\beta_t, \gamma_t, p_t, \alpha_t, \theta_t$ and $\eta_t$. The detailed proof of Theorem 2 is deferred to Appendix B.

**Theorem 2** *Suppose that Assumption 1 holds and the compression operators $\{\mathcal{C}_i^t\}$ used in Algorithm 1 satisfy (2) of Definition 2. Let $b = \min\left\{\omega, \sqrt{\frac{\omega(1+\omega)^2}{n}}\right\}$ and choose the two positive*

---

[3]Exception: While we indeed choose $\beta_t \equiv \beta$ for $t \geq 1$, the value of $\beta_0$ may be different.

*sequences $\{\beta_t\}$ and $\{\gamma_t\}$ as follows:*

$$\beta_t = \begin{cases} \beta_0 = \frac{9(1+b+\omega)^2}{(1+b)L} & \text{for } t = 0 \\ \beta \equiv \frac{48\omega(1+\omega)(1+b+2(1+\omega))}{n(1+b)^2} & \text{for } t \geq 1 \end{cases}, \qquad \gamma_t = \gamma \equiv \frac{(1+b)^2}{8(1+b+2(1+\omega))} \quad \text{for } t \geq 0. \tag{10}$$

*If we set the probabilities $\{p_t\}$ and positive stepsizes $\{\alpha_t\}, \{\eta_t\}, \{\theta_t\}$ of Algorithm 1 as follows:*

$$p_t \equiv \frac{1}{1+b}, \qquad \alpha_t \equiv \frac{1}{1+\omega}, \qquad \theta_t = \frac{3(1+b)}{t + 9(1+b+\omega)}, \quad \text{for } t \geq 0, \tag{11}$$

*and*

$$\eta_t = \begin{cases} \frac{1}{L(\beta_0 + 3/2)} & \text{for } t = 0 \\ \min\left\{ \left(1 + \frac{1}{t+9(1+b+\omega)}\right)\eta_{t-1}, \ \frac{1}{L(\beta+3/2)} \right\} & \text{for } t \geq 1 \end{cases}. \tag{12}$$

*Then* CANITA *(Algorithm 1) for all $T \geq 0$ satisfies*

$$\mathbb{E}\left[F^{T+1}\right] \leq O\left(\frac{(1 + \sqrt{\omega^3/n})L}{T^2} + \frac{\omega^3}{T^3}\right). \tag{13}$$

*According to* (13)*, the number of communication rounds for* CANITA *(Algorithm 1) to find an $\epsilon$-solution of the distributed problem* (1)*, i.e.,*

$$\mathbb{E}\left[f(w^{T+1}) - f(x^*)\right] \overset{(4)}{:=} \mathbb{E}\left[F^{T+1}\right] \leq \epsilon,$$

*is at most*

$$T = O\left(\sqrt{\left(1 + \sqrt{\frac{\omega^3}{n}}\right)\frac{L}{\epsilon}} + \omega\left(\frac{1}{\epsilon}\right)^{\frac{1}{3}}\right).$$

## 6 Experiments

In this section, we demonstrate the performance of our accelerated method CANITA (Algorithm 1) and previous methods QSGD and DIANA (the theoretical convergence results of these algorithms can be found in Table 1) with different compression operators on the logistic regression problem,

$$\min_{x \in \mathbb{R}^d} f(x) := \frac{1}{n}\sum_{i=1}^n \log\left(1 + \exp(-b_i a_i^T x)\right), \tag{14}$$

where $\{a_i, b_i\}_{i=1}^n \in \mathbb{R}^d \times \{\pm 1\}$ are data samples. We use three standard datasets: a9a, mushrooms, and w8a in the experiments. All datasets are downloaded from LIBSVM [4].

Similar to Li et al. [26], we also use three different compression operators: *random sparsification* (e.g. [39]), *natural compression* (e.g. [8]), and *random quantization* (e.g. [1]). In particular, we follow the same settings as in Li et al. [26]. For random-$r$ sparsification, the number of communicated bits per iteration is $32r$, and we choose $r = d/4$. For natural compression, the number of communicated bits per iteration is $9d$ bits [8]. For random $(2, s)$-quantization, we choose $s = \sqrt{d}$, which means the number of communicated bits per iteration is $2.8d + 32$ [1]. The default number of nodes/machines/workers is 20. In our experiments, we directly use the theoretical stepsizes and parameters for all three algorithms: QSGD [1, 24], DIANA [12], our CANITA (Algorithm 1). To compare with the settings of DIANA and CANITA, we use local gradients (not stochastic gradients) in QSGD. Thus here QSGD is equivalent to DC-GD provided in [24].

In Figures 1–3, we compare our CANITA with QSGD and DIANA with three compression operators: random sparsification (left), natural compression (middle), and random quantization (right) on three datasets: a9a (Figure 1), mushrooms (Figure 2), and w8a (Figure 3). The $x$-axis and $y$-axis represent the number of communication bits and the training loss, respectively.

Regarding the different compression operators, the experimental results indicate that natural compression and random quantization are better than random sparsification for all three algorithms. For

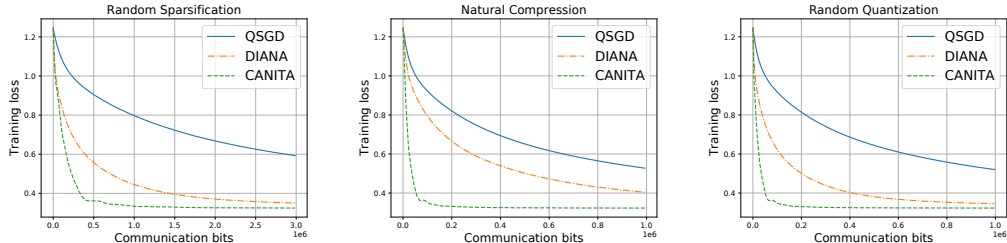

Figure 1: Performance of different methods for three different compressors (random sparsification, natural compression, and random quantization) on the `a9a` dataset.

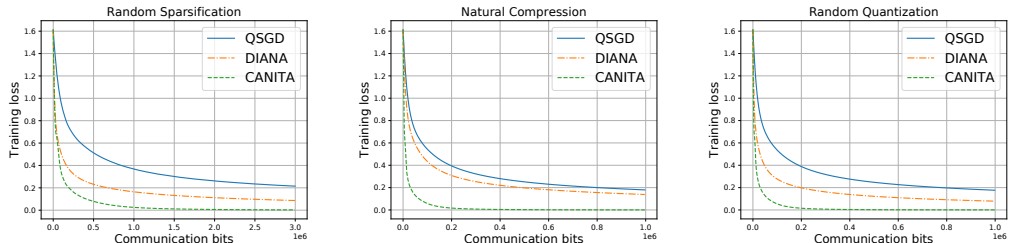

Figure 2: Performance of different methods for three different compressors (random sparsification, natural compression, and random quantization) on the `mushrooms` dataset.

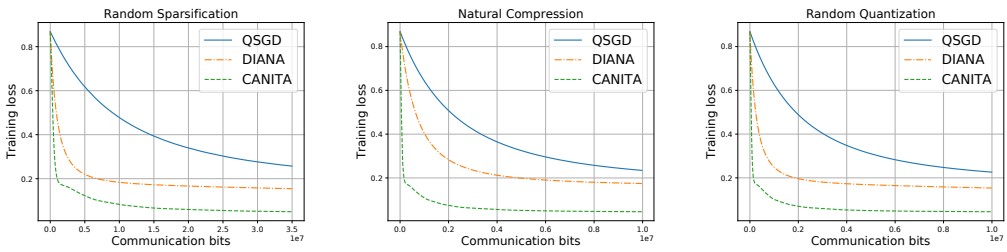

Figure 3: Performance of different methods for three different compressors (random sparsification, natural compression, and random quantization) on the `w8a` dataset.

instance, in Figure 1, DIANA uses $1.5 \times 10^6$ (random sparsification), $1.0 \times 10^6$ (natural compression), $0.4 \times 10^6$ (random quantization) communication bits for achieving the loss $0.4$, respectively.

Moreover, regarding the different algorithms, the experimental results indeed show that our CANITA converges the fastest compared with both QSGD and DIANA for all three compressors in all Figures 1–3, validating the theoretical results (see Table 1) and confirming the practical superiority of our accelerated CANITA method.

## 7 Conclusion

In this paper, we proposed CANITA: the first gradient method for distributed *general convex* optimization provably enjoying the benefits of both *communication compression* and *convergence acceleration*. There is very limited work on combing compression and acceleration. Indeed, previous works only focus on the (much simpler) strongly convex setting. We hope that our novel algorithm and analysis can provide new insights and shed light on future work in this line of research. We leave further improvements to future work. For example, one may ask whether our approach can be combined with the benefits provided by multiple local update steps [29, 38, 11, 10, 42], with additional variance reduction techniques [9, 24], and to what extent one can extend our results to structured nonconvex problems [22, 19, 27, 21, 25, 6, 35, 5].

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
