# Contents

# A Missing Proof for Theorem 1 in Section 5.1

In order to prove Theorem 1, we first formulate six auxiliary results (Lemmas 1–6) in Appendix A.1. The detailed proofs of these lemmas are deferred to Appendix C. Then in Appendix A.2 we show that Theorem 1 follows from Lemma 6.

## A.1 Six lemmas

First, we need a useful Lemma 1 which captures the change of the function value after a single gradient update step.

**Lemma 1** *Suppose that Assumption 1 holds. For any $\beta_t > 0$, the following equation holds for* CANITA *(Algorithm 1) for any round $t \geq 0$:*

$$\mathbb{E}\left[f(z^{t+1})\right] \leq \mathbb{E}\left[f(y^t) + \langle \nabla f(y^t), \theta_t(x^* - x^t)\rangle + \frac{\theta_t^2}{\eta_t}\left(D^t - D^{t+1}\right)\right.$$
$$\left. - \left(\frac{\theta_t^2}{2\eta_t} - \frac{L(1+\beta_t)\theta_t^2}{2}\right)\|x^{t+1} - x^t\|^2 + \frac{1}{2L\beta_t}\|\nabla f(y^t) - g^t\|^2\right]. \quad (15)$$

Note that

$$z^{t+1} - y^t = \theta_t(x^{t+1} - x^t) = -\eta_t g^t$$

according to the two momentum/interpolation steps of CANITA (see Line 3 and Line 10 of Algorithm 1) and the gradient update step (see Line 9 of Algorithm 1). The proof of Lemma 1 uses these relations and the smoothness Assumption 1.

In the next lemma, we bound the last variance term $\mathbb{E}\left[\|\nabla f(y^t) - g^t\|^2\right]$ appearing in (15) of Lemma 1. To simplify the notation, from now on we will write

$$Y^t := \frac{1}{n}\sum_{i=1}^{n}\|\nabla f_i(w^t) - \nabla f_i(y^t)\|^2, \quad (16)$$

and recall that $H^t := \frac{1}{n}\sum_{i=1}^{n}\|\nabla f_i(w^t) - h_i^t\|^2$ defined in (4).

**Lemma 2** *If $g^t$ is as defined in Line 8 of Algorithm 1, and the compression operator $\mathcal{C}_i^t$ satisfies* (2) *of Definition 2, we have*

$$\mathbb{E}\left[\|\nabla f(y^t) - g^t\|^2\right] \leq \frac{2\omega}{n}\left(Y^t + H^t\right). \quad (17)$$

This lemma is proved by using the definition of the $\omega$-compression operator (i.e., (2)).

Now, we need to bound the terms $Y^t$ and $H^t$ in (17) of Lemma 2. We first show how to handle the term $H^t$ in the following Lemma 3.

**Lemma 3** *Suppose that Assumption 1 holds and let $\alpha_t \leq \frac{1}{1+\omega}$. According to the probabilistic update of $w^{t+1}$ in Line 11 of Algorithm 1, we have*

$$\mathbb{E}\left[H^{t+1}\right] \leq \left(1 - \frac{\alpha_t}{2}\right)H^t + 2p_t\left(1 + \frac{2p_t}{\alpha_t}\right)Y^t + 2p_t L^2\theta_t^2\left(1 + \frac{2p_t}{\alpha_t}\right)\mathbb{E}\left[\|x^{t+1} - x^t\|^2\right]. \quad (18)$$

This lemma is proved by using the update of $w^{t+1}$ (Line 11 of Algorithm 1) and $h_i^{t+1}$ (Line 6 of Algorithm 1), the property of $\omega$-compression operator (i.e., (2)), and the smoothness Assumption 1.

To deal with the term $Y^t$ in Lemmas 2 and 3, we need the following result.

**Lemma 4** *Suppose that Assumption 1 holds. For any $y^t, w^t \in \mathbb{R}^d$, the following inequality holds:*

$$Y^t \leq 2L\left(f(w^t) - f(y^t) - \langle\nabla f(y^t), w^t - y^t\rangle\right). \quad (19)$$

The proof of this lemma directly follows from a standard result characterizing the $L$-smoothness of convex functions.

Finally, we also need a result connecting the function values $f(z^{t+1})$ in (15) of Lemma 1 and $f(w^{t+1})$ in (7) of Theorem 1 (recall that $F^{t+1} := f(w^{t+1}) - f(x^*)$ in (4)).

**Lemma 5** *According to the probabilistic update of $w^{t+1}$ in Line 11 of Algorithm 1, we have*

$$\mathbb{E}[f(w^{t+1})] = p_t \mathbb{E}[f(z^{t+1})] + (1 - p_t)\mathbb{E}[f(w^t)]. \tag{20}$$

Now, we combine Lemmas 1–5 to obtain our final key lemma, which describes the recursive form of the objective function value after a single round.

**Lemma 6** *Suppose that Assumption 1 holds and the compression operators $\{\mathcal{C}_i^t\}$ used in Algorithm 1 satisfy (2) of Definition 2. For any two positive sequences $\{\beta_t\}$ and $\{\gamma_t\}$ such that the probabilities $\{p_t\}$ and positive stepsizes $\{\alpha_t\}, \{\eta_t\}, \{\theta_t\}$ of Algorithm 1 satisfy the following relations*

$$\alpha_t \leq \frac{1}{1+\omega}, \qquad \eta_t \leq \frac{1}{L\left(1 + \beta_t + 4p_t\gamma_t\left(1 + \frac{2p_t}{\alpha_t}\right)\right)} \tag{21}$$

*for all $t \geq 0$, and*

$$\frac{2\omega}{\beta_t n} + 4p_t\gamma_t\left(1 + \frac{2p_t}{\alpha_t}\right) \leq 1 - \theta_t \tag{22}$$

*for all $t \geq 1$. Then the sequences $\{x^t, w^t, h_i^t\}$ of* CANITA *(Algorithm 1) for all $t \geq 0$ satisfy the inequality*

$$\mathbb{E}\left[F^{t+1} + \frac{\gamma_t p_t}{L}H^{t+1}\right] \leq \mathbb{E}\left[(1 - \theta_t p_t)F^t + \left(\frac{\omega}{\beta_t n} + \left(1 - \frac{\alpha_t}{2}\right)\gamma_t\right)\frac{p_t}{L}H^t + \frac{\theta_t^2 p_t}{\eta_t}\left(D^t - D^{t+1}\right)\right]. \tag{23}$$

## A.2 Proof of Theorem 1

Now, we are ready to prove the main convergence Theorem 1. According to Lemma 6, we know the change of the function value after each round. By dividing (23) with $\frac{\theta_t^2 p_t}{\eta_t}$ on both sides, we obtain

$$\mathbb{E}\left[\frac{\eta_t}{\theta_t^2 p_t}F^{t+1} + \frac{\gamma_t \eta_t}{\theta_t^2 L}H^{t+1}\right] \leq \mathbb{E}\left[\frac{(1 - \theta_t p_t)\eta_t}{\theta_t^2 p_t}F^t + \left(\frac{\omega}{\beta_t n} + \left(1 - \frac{\alpha_t}{2}\right)\gamma_t\right)\frac{\eta_t}{\theta_t^2 L}H^t + D^t - D^{t+1}\right]. \tag{24}$$

Then according to the following conditions on the parameters (see (6) of Theorem 1):

$$\frac{(1 - p_t\theta_t)\eta_t}{p_t\theta_t^2} \leq \frac{\eta_{t-1}}{p_{t-1}\theta_{t-1}^2}, \quad \text{and} \quad \left(\frac{\omega}{\beta_t n} + \left(1 - \frac{\alpha_t}{2}\right)\gamma_t\right)\frac{\eta_t}{\theta_t^2} \leq \frac{\gamma_{t-1}\eta_{t-1}}{\theta_{t-1}^2}, \quad \forall t \geq 1. \tag{25}$$

The proof of Theorem 1 is finished by telescoping (24) from $t = 1$ to $T$ via (25) and maintaining the same inequality (24) for $t = 0$:

$$\mathbb{E}\left[F^{T+1} + \frac{\gamma_T p_T}{L}H^{T+1}\right] \leq \frac{\theta_T^2 p_T}{\eta_T}\left(\frac{(1 - \theta_0 p_0)\eta_0}{\theta_0^2 p_0}F^0 + \left(\frac{\omega}{\beta_0 n} + \left(1 - \frac{\alpha_0}{2}\right)\gamma_0\right)\frac{\eta_0}{\theta_0^2 L}H^0 + D^0\right). \tag{26}$$

$\square$

# B Missing Proof for Theorem 2 in Section 5.2

In this appendix, we provide the proof for concrete Theorem 2 (which leads to a detailed convergence result). First, let us verify that the choice of parameters (i.e., (10)–(12)) in Theorem 2 satisfies the conditions (i.e., (5) and (6)) in Theorem 1. According to $p_t$ and $\alpha_t$ in (11) and $\gamma_t$ in (10), we have

$$4p_t\gamma_t\left(1 + \frac{2p_t}{\alpha_t}\right) = \frac{1}{2}, \quad \forall t \geq 0. \tag{27}$$

Then according to (27), $\eta_t$ of (12) and $\alpha_t$ of (11), the first two conditions in (5) of Theorem 1 are satisfied, i.e.,

$$\eta_t \leq \frac{1}{L\left(1 + \beta_t + 4p_t\gamma_t\left(1 + \frac{2p_t}{\alpha_t}\right)\right)} \quad \text{and} \quad \alpha_t \leq \frac{1}{1+\omega}, \quad \forall t \geq 0.$$

Besides, from (10) and (11), we know that $\theta_t \leq \frac{1}{3}$ and $\frac{2\omega}{\beta_t n} \leq \frac{1}{6}$ for any $t \geq 1$. Combining with (27), then the following condition in (6) of Theorem 1 is satisfied:

$$\frac{2\omega}{\beta_t n} + 4p_t\gamma_t\left(1 + \frac{2p_t}{\alpha_t}\right) \leq 1 - \theta_t, \quad \forall t \geq 1.$$

Now, only the following two conditions in (6) of Theorem 1 are remained:

$$\frac{(1 - p_t\theta_t)\eta_t}{p_t\theta_t^2} \leq \frac{\eta_{t-1}}{p_{t-1}\theta_{t-1}^2}, \quad \text{and} \quad \left(\frac{\omega}{\beta_t n} + \left(1 - \frac{\alpha_t}{2}\right)\gamma_t\right)\frac{\eta_t}{\theta_t^2} \leq \frac{\gamma_{t-1}\eta_{t-1}}{\theta_{t-1}^2}, \quad \forall t \geq 1. \tag{28}$$

For the first condition of (28), by plugging the parameter choice $\{p_t\}$ and $\{\theta_t\}$ of (11), it is sufficient to let

$$\left(1 - \frac{3}{t + 9(1 + b + \omega)}\right)\eta_t \leq \left(1 - \frac{1}{t + 9(1 + b + \omega)}\right)^2 \eta_{t-1}, \quad \forall t \geq 1. \tag{29}$$

For satisfying (29), it is sufficient to choose $\eta_t$ as in (12):

$$\eta_t = \min\left\{\left(1 + \frac{1}{t + 9(1 + b + \omega)}\right)\eta_{t-1}, \frac{1}{L(\beta + 3/2)}\right\}, \quad \forall t \geq 1. \tag{30}$$

Similarly, for the second condition of (28), by plugging the parameter choice $\{\theta_t\}$ and $\{\alpha_t\}$ of (11), it is sufficient to let

$$\left(\frac{\omega}{\beta_t n} + \left(1 - \frac{1}{2(1+\omega)}\right)\gamma_t\right)\eta_t \leq \gamma_{t-1}\eta_{t-1}\left(1 - \frac{1}{t + 9(1 + b + \omega)}\right)^2, \quad \forall t \geq 1. \tag{31}$$

By plugging $\{\beta_t\}$ and $\{\gamma_t\}$ of (10) into (31), we have

$$\left(1 - \frac{1}{3(1+\omega)}\right)\eta_t \leq \eta_{t-1}\left(1 - \frac{1}{t + 9(1 + b + \omega)}\right)^2, \quad \forall t \geq 1. \tag{32}$$

Note that the choice of $\eta_t$ in (30) also satisfies (32).

Now, we have verified that all conditions of Theorem 1 are satisfied with the parameter choice in Theorem 2. Next, we obtain the detailed convergence results of CANITA by using this choice of parameters. According to Theorem 1, we know that the following equation holds for any $T > 0$:

$$\mathbb{E}\left[F^{T+1} + \frac{\gamma_T p_T}{L}H^{T+1}\right] \leq \frac{\theta_T^2 p_T}{\eta_T}\left(\frac{(1 - \theta_0 p_0)\eta_0}{\theta_0^2 p_0}F^0 + \left(\frac{\omega}{\beta_0 n} + \left(1 - \frac{\alpha_0}{2}\right)\gamma_0\right)\frac{\eta_0}{\theta_0^2 L}H^0 + D^0\right). \tag{33}$$

According to (11), we have

$$\theta_T^2 p_T = \frac{9(1 + b)}{(T + 9(1 + b + \omega))^2}. \tag{34}$$

According to (30), we have

$$\eta_T = \min\left\{\frac{T + 9(1 + b + \omega)}{9(1 + b + \omega)}\eta_0, \ \frac{1}{L(\beta + 3/2)}\right\}$$

$$= \min\left\{\frac{T + 9(1 + b + \omega)}{9(1 + b + \omega)}\frac{1}{L(\beta_0 + 3/2)}, \ \frac{1}{L(\beta + 3/2)}\right\}$$

$$= \min\left\{\frac{(T + 9(1 + b + \omega))(1 + b)}{162(1 + b + \omega)^3}, \ \frac{1}{L(\beta + 3/2)}\right\}, \tag{35}$$

where (35) uses the appropriate $\beta_0 = \frac{9(1+b+\omega)^2}{(1+b)L}$ chosen in (10) of Theorem 2. Besides, according to the initial values of the parameters, we can simplify the right-hand-side of (33) with $\frac{(1-\theta_0 p_0)\eta_0}{\theta_0^2 p_0} \leq 1$ and $\left(1 - \frac{\alpha_0}{2}\right)\gamma_0\frac{\eta_0}{\theta_0^2 L} \leq 1$.

Now we plug (34) and (35) into (33) and omit the constant to obtain

$$\mathbb{E}\left[F^{T+1}\right] \leq O\left(\max\left\{\frac{(1 + b + \omega)^3}{(T + 9(1 + b + \omega))^3}, \ \frac{(1 + b)(\beta + 3/2)L}{(T + 9(1 + b + \omega))^2}\right\}\right)$$

$$\leq O\left(\max\left\{\frac{(1 + b + \omega)^3}{T^3}, \ \frac{(1 + b)(\beta + 3/2)L}{T^2}\right\}\right)$$

$$\leq O\left(\max\left\{\frac{(1 + \omega)^3}{T^3}, \ \frac{(1 + \sqrt{\omega(1 + \omega)^2/n})L}{T^2}\right\}\right) \tag{36}$$

$$= O\left(\frac{(1 + \sqrt{\omega^3/n})L}{T^2} + \frac{\omega^3}{T^3}\right), \tag{37}$$

where (36) uses $b = \min\left\{\omega, \sqrt{\frac{\omega(1+\omega)^2}{n}}\right\}$ and $\beta$ of (10) . Following from (37), we know that the number of communication rounds for CANITA (Algorithm 1) to find an $\epsilon$-solution such that

$$\mathbb{E}\left[f(w^{T+1}) - f(x^*)\right] \overset{(4)}{:=} \mathbb{E}\left[F^{T+1}\right] \leq \epsilon$$

is at most

$$T = O\left(\sqrt{\left(1 + \sqrt{\frac{\omega^3}{n}}\right)\frac{L}{\epsilon}} + \omega\left(\frac{1}{\epsilon}\right)^{\frac{1}{3}}\right).$$

$\square$

# C   Missing Proofs for Six Lemmas in Appendix A.1

In Appendix A, we provided the proof of Theorem 1 using six lemmas. Now we present the omitted proofs for these Lemmas 1–6 in Appendices C.1–C.6, respectively.

## C.1   Proof of Lemma 1

According to the $L$-smoothness of $f$ (Assumption 1), we have

$$
\begin{aligned}
&\mathbb{E}\left[f(z^{t+1})\right] \\
&\leq \mathbb{E}\left[f(y^t) + \langle \nabla f(y^t), z^{t+1} - y^t \rangle + \frac{L}{2}\|z^{t+1} - y^t\|^2\right] \\
&= \mathbb{E}\left[f(y^t) + \langle \nabla f(y^t), \theta_t(x^{t+1} - x^t) \rangle + \frac{L\theta_t^2}{2}\|x^{t+1} - x^t\|^2\right] \quad (38) \\
&= \mathbb{E}\left[f(y^t) + \langle \nabla f(y^t) - g^t, \theta_t(x^{t+1} - x^t) \rangle + \langle g^t, \theta_t(x^{t+1} - x^t) \rangle + \frac{L\theta_t^2}{2}\|x^{t+1} - x^t\|^2\right] \\
&\leq \mathbb{E}\left[f(y^t) + \frac{1}{2L\beta_t}\|\nabla f(y^t) - g^t\|^2 + \frac{L\beta_t\theta_t^2}{2}\|x^{t+1} - x^t\|^2 + \frac{L\theta_t^2}{2}\|x^{t+1} - x^t\|^2 \right. \\
&\qquad \left. + \langle g^t, \theta_t(x^{t+1} - x^t) \rangle\right] \quad (39) \\
&= \mathbb{E}\left[f(y^t) + \frac{1}{2L\beta_t}\|\nabla f(y^t) - g^t\|^2 + \frac{L(1+\beta_t)\theta_t^2}{2}\|x^{t+1} - x^t\|^2 \right. \\
&\qquad \left. + \langle g^t, \theta_t(x^* - x^t) \rangle + \langle g^t, \theta_t(x^{t+1} - x^*) \rangle\right] \\
&= \mathbb{E}\left[f(y^t) + \frac{1}{2L\beta_t}\|\nabla f(y^t) - g^t\|^2 + \frac{L(1+\beta_t)\theta_t^2}{2}\|x^{t+1} - x^t\|^2 + \langle \nabla f(y^t), \theta_t(x^* - x^t) \rangle \right. \\
&\qquad \left. + \langle g^t, \theta_t(x^{t+1} - x^*) \rangle\right] \quad (40) \\
&= \mathbb{E}\left[f(y^t) + \frac{1}{2L\beta_t}\|\nabla f(y^t) - g^t\|^2 + \frac{L(1+\beta_t)\theta_t^2}{2}\|x^{t+1} - x^t\|^2 + \langle \nabla f(y^t), \theta_t(x^* - x^t) \rangle \right. \\
&\qquad \left. + \frac{\theta_t^2}{\eta_t}\langle x^t - x^{t+1}, x^{t+1} - x^* \rangle\right] \quad (41) \\
&= \mathbb{E}\left[f(y^t) + \frac{1}{2L\beta_t}\|\nabla f(y^t) - g^t\|^2 + \frac{L(1+\beta_t)\theta_t^2}{2}\|x^{t+1} - x^t\|^2 + \langle \nabla f(y^t), \theta_t(x^* - x^t) \rangle \right. \\
&\qquad \left. + \frac{\theta_t^2}{2\eta_t}\left(\|x^t - x^*\|^2 - \|x^t - x^{t+1}\|^2 - \|x^{t+1} - x^*\|^2\right)\right] \\
&= \mathbb{E}\left[f(y^t) + \langle \nabla f(y^t), \theta_t(x^* - x^t) \rangle + \frac{\theta_t^2}{2\eta_t}\left(\|x^t - x^*\|^2 - \|x^{t+1} - x^*\|^2\right) \right. \\
&\qquad \left. - \left(\frac{\theta_t^2}{2\eta_t} - \frac{L(1+\beta_t)\theta_t^2}{2}\right)\|x^{t+1} - x^t\|^2 + \frac{1}{2L\beta_t}\|\nabla f(y^t) - g^t\|^2\right],
\end{aligned}
$$

where (38) holds since $z^{t+1} - y^t = \theta_t(x^{t+1} - x^t)$ according to the two momentum/interpolation steps of CANITA (see Line 3 and Line 10 of Algorithm 1), (39) uses Young's inequality with any $\beta_t > 0$, (40) holds due to $\mathbb{E}[g^t] = \nabla f(y^t)$ since the compression is unbiased from (2), and (41) holds according to the gradient update step $x^{t+1} = x^t - \frac{\eta_t}{\theta_t}g^t$ (see Line 9 of Algorithm 1).  $\square$

## C.2 Proof of Lemma 2

This lemma is proved as follows:

$$
\begin{aligned}
\mathbb{E}\left[\|\nabla f(y^t) - g^t\|^2\right] &= \mathbb{E}\left[\left\|\frac{1}{n}\sum_{i=1}^{n}\left(\mathcal{C}_i^t(\nabla f_i(y^t) - h_i^t) + h_i^t - \nabla f_i(y^t)\right)\right\|^2\right]\\
&= \frac{1}{n^2}\sum_{i=1}^{n}\mathbb{E}\left[\|\mathcal{C}_i^t(\nabla f_i(y^t) - h_i^t) + h_i^t - \nabla f_i(y^t)\|^2\right]\\
&\leq \frac{\omega}{n^2}\sum_{i=1}^{n}\|\nabla f_i(y^t) - h_i^t\|^2 \tag{42}\\
&\leq \frac{2\omega}{n^2}\sum_{i=1}^{n}\|\nabla f_i(y^t) - \nabla f_i(w^t)\|^2 + \frac{2\omega}{n^2}\sum_{i=1}^{n}\|\nabla f_i(w^t) - h_i^t\|^2, \tag{43}
\end{aligned}
$$

where (42) follows from the definition of $\omega$-compression operator (i.e., (2)), and the last inequality (43) uses Cauchy-Schwarz inequality. $\qquad\square$

## C.3 Proof of Lemma 3

Firstly, according to the probabilistic update of $w^{t+1}$ (see Line 11 of Algorithm 1) and recalling that $H^t := \frac{1}{n}\sum_{i=1}^{n}\|\nabla f_i(w^t) - h_i^t\|^2$ defined in (4), we get

$$
\mathbb{E}\left[H^{t+1}\right]
$$
$$
= \frac{p_t}{n}\sum_{i=1}^{n}\mathbb{E}\left[\|\nabla f_i(z^{t+1}) - h_i^{t+1}\|^2\right] + \frac{1-p_t}{n}\sum_{i=1}^{n}\mathbb{E}\left[\|\nabla f_i(w^t) - h_i^{t+1}\|^2\right]
$$
$$
\leq \left(1 + \frac{2p_t}{\alpha_t}\right)\frac{p_t}{n}\sum_{i=1}^{n}\mathbb{E}\left[\|\nabla f_i(z^{t+1}) - \nabla f_i(w^t)\|^2\right] + \left(1 + \frac{\alpha_t}{2p_t}\right)\frac{p_t}{n}\sum_{i=1}^{n}\mathbb{E}\left[\|\nabla f_i(w^t) - h_i^{t+1}\|^2\right]
$$
$$
\quad + \frac{1-p_t}{n}\sum_{i=1}^{n}\mathbb{E}\left[\|\nabla f_i(w^t) - h_i^{t+1}\|^2\right]. \tag{44}
$$
$$
\leq \left(1 + \frac{2p_t}{\alpha_t}\right)\frac{p_t}{n}\sum_{i=1}^{n}\mathbb{E}\left[\|\nabla f_i(z^{t+1}) - \nabla f_i(w^t)\|^2\right] + \left(1 + \frac{\alpha_t}{2}\right)\left(1 - 2\alpha_t + \alpha_t^2(1+\omega)\right)H^t \tag{45}
$$
$$
\leq \left(1 + \frac{2p_t}{\alpha_t}\right)\frac{p_t}{n}\sum_{i=1}^{n}\mathbb{E}\left[\|\nabla f_i(z^{t+1}) - \nabla f_i(w^t)\|^2\right] + \left(1 - \frac{\alpha_t}{2}\right)H^t \tag{46}
$$
$$
\leq \left(1 + \frac{2p_t}{\alpha_t}\right)\frac{2p_t}{n}\sum_{i=1}^{n}\mathbb{E}\left[\|\nabla f_i(z^{t+1}) - \nabla f_i(y^t)\|^2 + \|\nabla f_i(y^t) - \nabla f_i(w^t)\|^2\right] + \left(1 - \frac{\alpha_t}{2}\right)H^t \tag{47}
$$
$$
\leq \left(1 + \frac{2p_t}{\alpha_t}\right)\frac{2p_t}{n}\sum_{i=1}^{n}\mathbb{E}\left[L^2\|z^{t+1} - y^t\|^2 + \|\nabla f_i(y^t) - \nabla f_i(w^t)\|^2\right] + \left(1 - \frac{\alpha_t}{2}\right)H^t \tag{48}
$$
$$
\leq 2p_t L^2\theta_t^2\left(1 + \frac{2p_t}{\alpha_t}\right)\mathbb{E}\left[\|x^{t+1} - x^t\|^2\right] + 2p_t\left(1 + \frac{2p_t}{\alpha_t}\right)Y^t + \left(1 - \frac{\alpha_t}{2}\right)H^t, \tag{49}
$$

where (44) uses Young's inequality, (45) uses the update of local shifts $h_i^{t+1} = h_i^t + \alpha_t\mathcal{C}_i^t(\nabla f_i(w^t) - h_i^t)$ (see Line 6 of Algorithm 1) and the property of $\omega$-compression operator (i.e., (2)), (46) uses $\alpha_t \leq 1/(1+\omega)$, (47) uses Cauchy-Schwarz inequality, (48) uses the $L$-smoothness of $f_i$ (Assumption 1), and the last inequality (49) holds since $z^{t+1} - y^t = \theta_t(x^{t+1} - x^t)$ according to the two interpolation steps of CANITA (see Line 3 and Line 10 of Algorithm 1). $\qquad\square$

## C.4 Proof of Lemma 4

This lemma directly follows from a standard result under Assumption 1. According to e.g. Lemma 1 of [18] or Lemma 5 of [20], we have

$$\frac{1}{2L}\|\nabla f_i(w^t) - \nabla f_i(y^t)\|^2 \le f_i(w^t) - f_i(y^t) - \langle \nabla f_i(y^t), w^t - y^t \rangle. \tag{50}$$

Then, the result (19) is obtained by summing up (50) for all $i \in [n]$ and noting $f(x) := \frac{1}{n}\sum_{i=1}^n f_i(x)$ (see (1)) and $Y^t := \frac{1}{n}\sum_{i=1}^n \|\nabla f_i(w^t) - \nabla f_i(y^t)\|^2$ (see (16)). $\qquad\square$

## C.5 Proof of Lemma 5

The lemma follows directly from the probabilistic update of $w^{t+1}$; see Line 11 of Algorithm 1. $\quad\square$

## C.6 Proof of Lemma 6

Now, we provide the detailed proof for the key Lemma 6 by using previous Lemmas 1–5. First, we plug (17) of Lemma 2 into (15) of Lemma 1 to obtain

$$\mathbb{E}\left[f(z^{t+1})\right] \le \mathbb{E}\left[f(y^t) + \langle \nabla f(y^t), \theta_t(x^* - x^t)\rangle + \frac{\theta_t^2}{\eta_t}\left(D^t - D^{t+1}\right)\right.$$
$$\left. - \left(\frac{\theta_t^2}{2\eta_t} - \frac{L(1+\beta_t)\theta_t^2}{2}\right)\|x^{t+1} - x^t\|^2 + \frac{\omega}{L\beta_t n}Y^t + \frac{\omega}{L\beta_t n}H^t\right]. \tag{51}$$

Then, we add (51) and $\frac{\gamma_t}{L} \times$ (18) of Lemma 3 to get

$$\mathbb{E}\left[f(z^{t+1}) + \frac{\gamma_t}{L}H^{t+1}\right]$$

$$\le \mathbb{E}\left[f(y^t) + \langle \nabla f(y^t), \theta_t(x^* - x^t)\rangle + \frac{\theta_t^2}{\eta_t}\left(D^t - D^{t+1}\right)\right.$$
$$- \left(\frac{\theta_t^2}{2\eta_t} - \frac{L(1+\beta_t)\theta_t^2}{2}\right)\|x^{t+1} - x^t\|^2 + \frac{\omega}{L\beta_t n}Y^t + \frac{\omega}{L\beta_t n}H^t$$
$$\left. + \left(1 - \frac{\alpha_t}{2}\right)\frac{\gamma_t}{L}H^t + \left(1 + \frac{2p_t}{\alpha_t}\right)\frac{2p_t\gamma_t}{L}Y^t + 2p_t\gamma_t L\theta_t^2\left(1 + \frac{2p_t}{\alpha_t}\right)\|x^{t+1} - x^t\|^2\right]$$

$$= \mathbb{E}\left[f(y^t) + \langle \nabla f(y^t), \theta_t(x^* - x^t)\rangle + \frac{\theta_t^2}{\eta_t}\left(D^t - D^{t+1}\right)\right.$$
$$- \left(\frac{\theta_t^2}{2\eta_t} - \frac{L(1+\beta_t)\theta_t^2}{2} - 2p_t\gamma_t L\theta_t^2\left(1 + \frac{2p_t}{\alpha_t}\right)\right)\|x^{t+1} - x^t\|^2$$
$$\left. + \left(\frac{\omega}{\beta_t n} + \left(1 - \frac{\alpha_t}{2}\right)\gamma_t\right)\frac{1}{L}H^t + \left(\frac{2\omega}{\beta_t n} + 4p_t\gamma_t\left(1 + \frac{2p_t}{\alpha_t}\right)\right)\frac{1}{2L}Y^t\right]$$

$$\le \mathbb{E}\left[f(y^t) + \langle \nabla f(y^t), \theta_t(x^* - x^t)\rangle + \frac{\theta_t^2}{\eta_t}\left(D^t - D^{t+1}\right) + \left(\frac{\omega}{\beta_t n} + \left(1 - \frac{\alpha_t}{2}\right)\gamma_t\right)\frac{1}{L}H^t\right.$$
$$\left. + \left(\frac{2\omega}{\beta_t n} + 4p_t\gamma_t\left(1 + \frac{2p_t}{\alpha_t}\right)\right)\frac{1}{2L}Y^t\right] \tag{52}$$

$$\le \mathbb{E}\left[f(y^t) + \langle \nabla f(y^t), \theta_t(x^* - x^t)\rangle + \frac{\theta_t^2}{\eta_t}\left(D^t - D^{t+1}\right) + \left(\frac{\omega}{\beta_t n} + \left(1 - \frac{\alpha_t}{2}\right)\gamma_t\right)\frac{1}{L}H^t\right.$$
$$\left. + \frac{1 - \theta_t}{2L}Y^t\right] \tag{53}$$

$$\le \mathbb{E}\left[f(y^t) + \langle \nabla f(y^t), \theta_t(x^* - x^t)\rangle + \frac{\theta_t^2}{\eta_t}\left(D^t - D^{t+1}\right) + \left(\frac{\omega}{\beta_t n} + \left(1 - \frac{\alpha_t}{2}\right)\gamma_t\right)\frac{1}{L}H^t\right.$$
$$\left. + (1 - \theta_t)\left(f(w^t) - f(y^t) - \langle \nabla f(y^t), w^t - y^t\rangle\right)\right] \tag{54}$$

$$= \mathbb{E}\left[f(y^t) + \langle \nabla f(y^t), \theta_t(x^* - x^t)\rangle + \frac{\theta_t^2}{\eta_t}\left(D^t - D^{t+1}\right) + \left(\frac{\omega}{\beta_t n} + \left(1 - \frac{\alpha_t}{2}\right)\gamma_t\right)\frac{1}{L}H^t \right.$$

$$\left. + (1 - \theta_t)\left(f(w^t) - f(y^t)\right) - \theta_t\langle \nabla f(y^t), y^t - x^t\rangle\right] \tag{55}$$

$$\leq \mathbb{E}\left[(1 - \theta_t)f(w^t) + \theta_t f(x^*) + \frac{\theta_t^2}{\eta_t}\left(D^t - D^{t+1}\right) + \left(\frac{\omega}{\beta_t n} + \left(1 - \frac{\alpha_t}{2}\right)\gamma_t\right)\frac{1}{L}H^t\right], \tag{56}$$

where (52) holds by letting $\eta_t \leq \frac{1}{L\left(1+\beta_t+4p_t\gamma_t(1+2p_t/\alpha_t)\right)}$, (53) holds by letting $\frac{2\omega}{\beta_t n} + 4p_t\gamma_t(1 + \frac{2p_t}{\alpha_t}) \leq 1 - \theta_t$, (54) follows from (19) of Lemma 4, (55) holds since $y^t = \theta_t x^t + (1 - \theta_t)w^t$ (see Line 3 of Algorithm 1), and the last inequality (56) uses the convexity of $f$. Also note that (53) from (52) uses $\frac{2\omega}{\beta_t n} + 4p_t\gamma_t(1 + \frac{2p_t}{\alpha_t}) \leq 1 - \theta_t$, however this condition is only needed for $t \geq 1$, i.e., it is not needed for the case $t = 0$ since $Y^0 = 0$ from $y^0 = w^0 = x^0$. The function and inner product terms will also perform the same result in the final (56) since $y^0 = w^0 = x^0$.

The proof of Lemma 6 is finished by adding (56) $\times p_t$ and (20) of Lemma 5 to obtain (23). $\qquad\square$