# OpenReview forum: "CANITA: Faster Rates for Distributed Convex Optimization with Communication Compression"
_NeurIPS.cc/2021/Conference — NeurIPS 2021 Poster_

### Official Review · Reviewer_y4iy · 2021-07-16

**Rating:** 7
**Confidence:** 4

**Summary:**

This paper studies the compressed and accelerated gradient method for nonstrongly convex federated learning. State-of-the-art accelerated convergence rate is established.

**Limitations And Societal Impact:**

No limitation is discussed. I suggest to include the numerical expriment in the final version.

**Main Review:**

Originality: This paper combines the accelerated ANITA in [16] and the compressed DIANA in [23]. I think it is a novel combination of two well-known techniques. It also extends the accelerated and compressed gradient method in [19] from strongly convex problems to nonstrongly convex ones. The convergence rate improvement over previous results is compared clearly.

Quality: The theory is technically solid. State-of-the-art convergence rate for nonstrongly convex problems is proved.

Clarity: This paper is written well. The proof is organized very clearly.

Significance: This paper studies acceleration and compression in federated learning. This is a significant topic. State-of-the-art convergence rate for nonstrongly convex problems is proved. The previous work on acceleration and compression all focus on strongly convex problems. I think the result is significant.

Major comment:

The experiment is missing. I suggest to include the numerical experiment in the final version. This paper proposes a new method, rather than reanalyzing a widely used method. So I think the numerical experiment is necessary.

Minor comment:

1. This paper studies the nonstrongly convex problems while [19] studied strongly convex ones. I am interested how fast CANITA converges for strongly convex problems by adoptting the proof techniques used in this paper. Is it faster than the one in [19]? The two rates in the last two lines of Table 1 looks different by letting $\kappa=\frac{L}{\epsilon}$ in the first one. I suggest to include the proof for strongly convex problmes in the supplementary material such that other researchers are likely to use the proof techniques in their work. For example,  the authors may organize the proof in a unified framework for both strongly convex and nonstrongly convex problems.

2. I suggest to remove the proof of Lemma 6 to the supplementary material and inlcude the numurical experiment in page 8.

3. The proof of Theorem 1 in page 16 is redundant, because it is proved in page 9.


**Time Spent Reviewing:**

6 hours

---

> ### Author Response · Authors · 2021-08-05
> **Main issue: Experiments**
>
> Issue: "The experiment is missing. I suggest to include the numerical experiment in the final version. This paper proposes a new method, rather than reanalyzing a widely used method. So I think the numerical experiment is necessary."
>
> Reply:
> - Thanks for this constructive suggestion!
> - We have already run some initial experiments on logistic regression problems, in which we compared our CANITA method with QSGD and DIANA. The numerical results indeed show that CANITA converges faster with less communication cost. We will add these and other experiments in the final version of the paper.

---

> > ### Comment · Reviewer_y4iy · 2021-08-25
> > **Reply to the rebuttal**
> >
> > Thanks for the rebuttal. I keep my score unchanged.

---

> ### Author Response · Authors · 2021-08-05
> **Minor comment: strongly convex case**
>
> Issue: "This paper studies the nonstrongly convex problems while [19] studied strongly convex ones. I am interested how fast CANITA converges for strongly convex problems by adoptting the proof techniques used in this paper. Is it faster than the one in [19]? The two rates in the last two lines of Table 1 looks different by letting $\kappa=\frac{L}{\epsilon}$ in the first one. I suggest to include the proof for strongly convex problems in the supplementary material such that other researchers are likely to use the proof techniques in their work. For example, the authors may organize the proof in a unified framework for both strongly convex and nonstrongly convex problems."
>
> Reply:
> - Thanks for this thoughtful comment!
> - We would like to refer the reviewer to our response to "Reviewer CD9C" related to "Question: strongly convex case".
> - Currently, we do not know whether our CANITA method is faster than the one in [19] in the strongly convex setting since we did not analyze CANITA in that setting.
> - Your observation of letting $\kappa=L/\epsilon$ in the first one makes some sense. Although the two rates look different, the order seems to be similar (if you move $\omega$ into the square root in the first one, you will get the same term $\sqrt{\omega^3/n}$. As you suggested, we will try to include the proof for the strongly convex case, but this may be difficult. If we succeed, we will also try to organize a unified proof for both convex and strongly convex problems in the final version, such that other researchers are more likely to use our proof techniques.

---

> ### Author Response · Authors · 2021-08-05
> **Minor comment: reorganization of some results**
>
> Issue: "I suggest to remove the proof of Lemma 6 to the supplementary material and include the numerical experiment in page 8. The proof of Theorem 1 in page 16 is redundant, because it is proved in page 9."
>
> Reply:
>
> - Following your suggestion, we will move the proof of Lemma 6 to the appendix.
> - We will add the experiments where you suggest.
> - We will remove the redundant proof of Theorem 1 in the final version of the paper.

---

> ### Author Response · Authors · 2021-08-06
> **Thanks!**
>
> Thanks for the positive evaluation of our paper and for valuing its originality, quality, clarity, and significance in detail!

---

### Official Review · Reviewer_CD9C · 2021-07-16

**Rating:** 7
**Confidence:** 3

**Summary:**

The paper propose a compressed and accelerated gradient method called CANITA for distributed optimization.
It improves the the state-of-the-art result which is achieved by DIANA in smooth and convex problems.
The main contribution is the (near) optimal algorithm and its theoretical analysis.

**Limitations And Societal Impact:**

Some questions:
1. Though CANITA has a theoretical superiority, it is still unclear how it performs empirically. I suggest the author could add some numerical experiments to investigate the empirical performance and its dependence on the hyper parameters.
2. I am curious about why not provide the convergence rate of CANITA for strongly convex cases? ANITA could handle convex and strongly convex cases simultaneously. So I naturally expect CANITA could achieve it also. Comparing ANITA and CANITA, I find that the key is how to perform update rule (line 9). So why not use $x_{t+1}=\frac{1}{1+\mu \eta_{t}}\left(x_{t}+\mu \eta_{t} \underline{x}_{t}\right)-\frac{\eta_{t}}{\theta_{t}} \widetilde{\nabla}_{t}$ in the line 9 of CANITA? Does there exist some fundamental difficulties？
3. A typo: In line 148, I guess it should be $\theta_t = \frac{a_3}{t+a_4}$.

**Main Review:**

Originality:

The paper considers a unexplored topic, how to fuse acceleration technique and unbiased compression operator organically.
Though the algorithm (CANITA) is a combination accelerated ANITA method and compressed DIANA method, the analysis of the resulting algorithm has its own difficulty, i.e., how to analyze the error introduced by the compression operator while not ruining the delicate structure of acceleration for convex problems.
I think the analysis is new.

Quality:

The theoretical analysis is valid.
The author also provides a proof sketch to illustrate how to construct a potential function for convergence analysis.

Clarity:

The paper is well written and clear.

Significance:

The paper improves the state-of-the-art result of non-accelerated compressed algorithms.
I think the work might give insights on how to combine compression and acceleration optimally.


======================================================


I have read the author's response.
I think the author well answers my questions.
I keep my score.


**Time Spent Reviewing:**

4

---

> ### Author Response · Authors · 2021-08-05
> **Thanks!**
>
> Thanks for the positive evaluation of our paper and for valuing its originality, quality, clarity, and significance in detail!

---

> ### Author Response · Authors · 2021-08-05
> **Question: experiments**
>
> Issue: "Though CANITA has a theoretical superiority, it is still unclear how it performs empirically. I suggest the author could add some numerical experiments to investigate the empirical performance and its dependence on the hyper parameters."
>
> Reply:
> - Thanks for this constructive suggestion.
> - We have already run some initial experiments on logistic regression problems, in which we compared our CANITA method with QSGD and DIANA. The numerical results indeed show that our CANITA method converges faster and with less communication cost. We will add the experiments in the final camera-ready version of the paper.

---

> ### Author Response · Authors · 2021-08-05
> **Question: strongly convex case**
>
> Issue: "I am curious about why not provide the convergence rate of CANITA for strongly convex cases? ANITA could handle convex and strongly convex cases simultaneously. So I naturally expect CANITA could achieve it also. Comparing ANITA and CANITA, I find that the key is how to perform update rule (line 9). So why not use $x_{t+1}=\frac{1}{1+\mu \eta_{t}}\left(x_{t}+\mu \eta_{t} \underline{x_t}\right)-\frac{\eta_{t}}{\theta_{t}} \widetilde{\nabla}_{t}$ in the line 9 of CANITA? Does there exist some fundamental difficulties"
>
> Reply:
> - Our initial thinking was as follows: the ADIANA method authors [19] already analyzed the strongly convex case and obtained acceleration, as we show in Table 1. However, there is no work on the general non-strongly convex case. So, in order to focus on what is not known, we omitted the more restrictive strongly convex case and tried to resolve the open problem: whether the benefits of compression and acceleration can be combined in the more general convex case as well.
> - Besides, your observation is correct. The key point is the update rule in line 9 for dealing with both convex and strongly convex cases simultaneously. If we use the similar update in ANITA for $x^{t+1}$ in line 9 of CANITA ,as you pointed out, we believe that CANITA can also deal with the strongly convex case as well. We do not think there exist some fundamental difficulties here.
> - Thanks for pointing out this point and letting us rethink it. We will think about this carefully and if we succeed, we will add the strongly convex case of CANITA later.

---

> ### Author Response · Authors · 2021-08-05
> **Question: typo on line 148**
>
> Issue: "A typo: In line 148, I guess it should be $\theta_t = \frac{a_3}{t+a_4}.$"
>
> Reply:
> - You are right. Thanks for pointing this out! We will correct.

---

### Official Review · Reviewer_Jr2k · 2021-07-16

**Rating:** 6
**Confidence:** 2

**Summary:**

This paper proposes a novel distributed methods with acceleration and communication compression. Authors proof the convergence and show that it is faster than existing SOTA methods.

**Limitations And Societal Impact:**

See above.

**Main Review:**

Positive:
This method achieves bigO(1/T^2) convergence rate benefit from the acceleration technique.

Negative:
1. I do not think this method is suitable for traditional distributed optimization problems or federated optimization problems. In traditional distributed optimization problems, the data size and dimension are large. Calculating \nabla f_i(w) is slow and usually is worse than calculating a stochastic gradient, even when the model is convex. Furthermore, additional parameters h_t, \bar{x}_t and w_t will cost huge memory. In federated  learning, usually only few machines participate training in each communication round.

2. Authors should add experiments to compare CANITA with other distributed communication efficient optimization methods. I think a convex optimization experiment can be easily conduct on pytorch and tensorflow.

**Time Spent Reviewing:**

2 hours

---

> ### Author Response · Authors · 2021-08-05
> **Issue: suitability for traditional distributed/federated optimization**
>
> Issue: "I do not think this method is suitable for traditional distributed optimization problems or federated optimization problems. In traditional distributed optimization problems, the data size and dimension are large. Calculating $\nabla f_i(w)$ is slow and usually is worse than calculating a stochastic gradient, even when the model is convex. Furthermore, additional parameters $h_t$, $\bar{x}_t$ and $w_t$ will cost huge memory. In federated learning, usually only few machines participate training in each communication round."
>
> Reply:
>
> - Calculation of gradients: We already replied to "Reviewer Yf7V" about the same issue in detail. Please can we refer you to that reply?
> - Parameters: We point out that in all methods, nodes always require the memory for at least one parameter: the model $x$ we are learning. In CANITA, the additional parameters are *reused* in memory across all iterations, i.e., the memory space does not accumulate, we only need one space for each parameter. Since we have three parameters, CANITA requires at most three times the standard memory cost of the least memory expensive methods. Moreover, it turns out that no accelerated methods or variance-reduced (shifted) compression methods can avoid introducing additional parameters, and CANITA is no exception. Thus, we do not think this additional memory is an issue; this is to be expected. Our improved theoretical rates are worth such a slim memory overhead!

---

> ### Author Response · Authors · 2021-08-05
> **Issue: partial participation**
>
> Issue: "In federated learning, usually only few machines participate training in each communication round."
>
> Reply:
> - Yes, we know that in cross-device federated learning (FL) partial participation is crucial. However, in cross-silo FL, partial participation is not needed, and this is the focus of our work (note we work with a finite number of nodes, and not with a population of nodes, as is often done in cross-device FL).
> - Moreover, many federated learning works only theoretically analyze the full participation setting, like we do. This is often done because the main difficulty of such works lies elsewhere. For example, many Local SGD papers ignore to deal with partial participation because the combination of local steps and SGD alone is already theoretically difficult and not understood, especially in the heterogeneous data regime. Likewise, in our regime, the difficult part is to combine acceleration and communication compression in the convex regime as there are no results in this realm. We argue that one first needs to obtain such results in such an idealized scenario before further extensions are to be investigated. Partial participation is such an extension, and should be dealt with in subsequent work(s). Replacing full gradients by inexact/stochastic gradients is another possible future extension that needs to be carefully investigated. We can think of many possible future avenues for research, but note that none of them would be possible without the groundwork that our paper provides.

---

> ### Author Response · Authors · 2021-08-05
> **Issue: experiments**
>
> Issue: "Authors should add experiments to compare CANITA with other distributed communication efficient optimization methods. I think a convex optimization experiment can be easily conduct on pytorch and tensorflow."
>
> Reply:
> - Thanks for this constructive suggestion. And thanks for not penalizing our work for lack of experiments which means you value theory!
> - We have already run some initial experiments on logistic regression tasks, in which we compared our CANITA method with QSGD and DIANA. The numerical results indeed show that our CANITA method converges faster and with less communication cost. We will add these and other experiments in the final version of the paper.

---

> ### Author Response · Authors · 2021-08-06
> **Thanks!**
>
> Thanks for your positive and helpful comments and for valuing our paper for:
> - proposing a novel distributed method with acceleration and communication compression,
> - achieving the accelerated convergence rate which is faster than previous theoretical SOTA.

---

### Official Review · Reviewer_Yf7V · 2021-07-18

**Rating:** 6
**Confidence:** 3

**Summary:**

This paper considers a distributed GD setting where multiple nodes need to jointly minimize the average of individual convex objectives by computing and communicating its compressed local gradients.  The proposed algorithm is shown to achieves an accelerated convergence rate with compressed gradient passing.

**Limitations And Societal Impact:**

Yes

**Main Review:**

This paper is clearly written and well organized. The proven convergence with the proposed algorithm is faster than a few earlier works.

My major concern is the usefulness of the studied setting. Requiring each node to compute its gradient (instead of stochastic/sampled gradient) in a distributed optimization does not seem practical. The setting is much less challenging than the conventional distributed (compressed) SGD setting as considered in QSGD in [1].  So it is unfair to say the results in the current paper improve [1] as in table 1.  The order-wise faster convergence (compared to [1]) is not quite surprising to me under this distributed GD setting.

**Time Spent Reviewing:**

2

---

> ### Author Response · Authors · 2021-08-05
> **Thanks!**
>
> Thanks for your positive and thoughtful comments and for valuing our paper for:
> - being clearly written and well organized,
> - proving faster convergence rates (using a new method) than previous theoretical SOTA.

---

> ### Author Response · Authors · 2021-08-05
> **Major concern: gradients vs stochastic gradients**
>
> Issue: "My major concern is the usefulness of the studied setting. Requiring each node to compute its gradient (instead of stochastic/sampled gradient) in a distributed optimization does not seem practical. The setting is much less challenging than the conventional distributed (compressed) SGD setting as considered in QSGD in [1]. So it is unfair to say the results in the current paper improve [1] as in table 1. The order-wise faster convergence (compared to [1]) is not quite surprising to me under this distributed GD setting."
>
> Reply:
> - The setting studied in our paper *is* useful in practice. Indeed, relying on exact/full gradient computation is both possible and desirable in the regime when the bottleneck of any algorithm/system is high cost of communication of messages between the workers and the master and not anything else, such as the computation cost on each node/machine/worker. In particular, if the data on each node is not very large (a typical scenario in federated learning), and/or if each node has large enough computational power (a typical scenario in commodity clusters), then the nodes are able to compute their gradients at a cost that can be far smaller than the cost of communication. This is the key scenario we focus in; and in this scenario CANITA works best.
> - We appreciate that you point out the possibility to study the stochastic setting, and we agree that this is an interesting subject of further study. However, it is also not trivial, and we made a conscious decision to leave this to future research. Indeed, note that we explicitly mention the stochastic setting (stochastic gradient, variance-reduced stochastic gradient [6,18]) as possible future work in our conclusion (Section 6).
> - Please note that no method works best in all scenarios. For example, if communication cost is not an issue, then communication compression is not necessary, and no method in this vast subarea of distributed optimization is relevant. This does not invalidate work in the area. It just means that the sweet spot/applicability domain for such method is elsewhere: in regime when the cost of communication is sufficiently high compared to other costs.
> - We disagree with the claim that the QSGD [1] setting is much more challenging if you are trying to say that the analysis of a method (such as QSGD) that uses i) communication compression and ii) subsampling is more challenging than analyzing a method (such as our method CANITA) that uses i) communication compression and iii) momentum/acceleration. In fact, we believe that the analysis of CANITA is much more involved, and the fact that there is no accelerated theoretical result in this area is an empirical evidence for this claim. In general, convergence/complexity proofs of accelerated/momentum methods are much more difficult than proofs convergence/complexity proofs of non-accelerated methods.
> - We agree with the claim that the QSGD [1] setting is more challenging if by this you mean a setting where, in addition to what we consider, one is unable to compute full gradients on each node, as we do. This is clearly true, almost tautologically, since the former imparts one more constraint on algorithms. However, this setting is not *much* more difficult in the analysis. Indeed, we believe that one could analyze CANITA with stochastic gradients as well. This can be considered as future work as we discussed above.
> - On inclusion of QSGD in Table 1: We include this in the table for several reasons. First, virtually everyone familiar with methods supporting compressed communication is familiar with QSGD just like everyone familiar with stochastic approximation is familiar with SGD. So, including theoretical result for QSGD is helpful didactically, and in order to put a flag at a traditional and well-known baseline. Further, QSGD *can* be applied as "QGD", that is, without stochastic gradients, and to distributed convex problems. Naturally, its rate will be slightly better (but still not accelerated) in this regime than in stochastic regime. So, it is a method that can solve the problem we propose, and hence is a completely valid method to compare to. Indeed, it is not surprising that other newer methods beat its rate - it would be surprising if they did not, as that would mean that progress in the area stalled a long time ago! This is not supposed to be surprising - it is supposed to be educational, and also scholarly, in that we give tribute to an early method in the area. Indeed, we are thankful to QSGD authors as they inspired our work (not only in this paper!).

---

### Author Response · Authors · 2021-08-16
**To All Reviewers**

Dear Reviewers,

Thanks for the effort you put into reading and commenting on our work, and for the positive evaluation of our paper. You have given our work scores: 6, 6, 7 and 7, and have described our paper positively in the following way:

- This paper is clearly written and well organized.
- The proven convergence with the proposed algorithm is faster than a few earlier works.
- This method achieves big O(1/T^2) convergence rate benefit from the acceleration technique.
- The paper considers a unexplored topic, how to fuse acceleration technique and unbiased compression operator organically.
- Though the algorithm (CANITA) is a combination accelerated ANITA method and compressed DIANA method, the analysis of the resulting algorithm has its own difficulty, i.e., how to analyze the error introduced by the compression operator while not ruining the delicate structure of acceleration for convex problems. I think the analysis is new.
- The theoretical analysis is valid. The author also provides a proof sketch to illustrate how to construct a potential function for convergence analysis.
- The paper is well written and clear.
- The paper improves the state-of-the-art result of non-accelerated compressed algorithms. I think the work might give insights on how to combine compression and acceleration optimally.
- This paper combines the accelerated ANITA in [16] and the compressed DIANA in [23]. I think it is a novel combination of two well-known techniques. It also extends the accelerated and compressed gradient method in [19] from strongly convex problems to nonstrongly convex ones. The convergence rate improvement over previous results is compared clearly.
- The theory is technically solid. State-of-the-art convergence rate for nonstrongly convex problems is proved.
- This paper is written well. The proof is organized very clearly.
- This paper studies acceleration and compression in federated learning. This is a significant topic.
- State-of-the-art convergence rate for nonstrongly convex problems is proved. The previous work on acceleration and compression all focus on strongly convex problems. I think the result is significant.

You have also raised some questions/concerns, to which we have replied in our detailed rebuttal. **Please can you let us know how well we did? Did we resolve your questions/concerns? If yes, we expect that you consider raising your scores appropriately. If not, what concerns remained unaddressed? We will be glad to get a chance to explain in more detail.**

Thanks for your time and support!

Authors

---

### Decision · Program_Chairs · 2021-09-27

**Decision:**

Accept (Poster)

**Comment:**

The review committee agreed that the paper is an interesting contribution and hence should be accepted for presentation at Neurips. However, there were some concerns regarding the practical relevance of the work. The authors mentioned in their response that they have run experiments on "logistic regression tasks". I strongly encourage the authors to include those simulations in the final version of the paper.